# MoPe 😔:
# Model Perturbation-based Privacy Attacks on Language Models

**Marvin Li**[*]
Harvard College

**Jason Wang** [*]
Harvard College

**Jeffrey Wang** [*]
Harvard College

**Seth Neel**[†]
Harvard University

## Abstract

Recent work has shown that Large Language Models (LLMs) can unintentionally leak sensitive information present in their training data. In this paper, we present $\text{MoPe}_\theta$ (**Mo**del **Pe**rturbations), a new method to identify with high confidence if a given text is in the training data of a pre-trained language model, given white-box access to the models parameters. $\text{MoPe}_\theta$ adds noise to the model in parameter space and measures the drop in log-likelihood at a given point $x$, a statistic we show approximates the trace of the Hessian matrix with respect to model parameters. Across language models ranging from 70M to 12B parameters, we show that $\text{MoPe}_\theta$ is more effective than existing loss-based attacks and recently proposed perturbation-based methods. We also examine the role of training point order and model size in attack success, and empirically demonstrate that $\text{MoPe}_\theta$ accurately approximate the trace of the Hessian in practice. Our results show that the loss of a point alone is insufficient to determine extractability—there are training points we can recover using our method that have average loss. This casts some doubt on prior works that use the loss of a point as evidence of memorization or "unlearning."

## 1 Introduction

Over the last few years, Large Language Models or LLMs have set new standards in performance across a range of tasks in natural language understanding and generation, often with very limited supervision on the task at hand (Brown et al., 2020). As a result, opportunities to use these models in real-world applications proliferate, and companies are rushing to deploy them in applications as diverse as A.I. assisted clinical diagnoses (Sharma et al., 2023), NLP tasks in finance (Wu et al., 2023), or an A.I. "love coach"

(Soper, 2023). While early state of the art LLMs have been largely trained on public web data (Radford et al., 2019; Gao et al., 2021; Biderman et al., 2023b; Black et al., 2021), increasingly models are being fine-tuned on data more relevant to their intended domain, or even trained from scratch on this domain specific data. In addition to increased performance on range of tasks, training models from scratch is attractive to companies because early work has shown it can mitigate some of the undesirable behavior associated with LLMs such as hallucination (Ji et al., 2022), toxicity (Gehman et al., 2020), as well as copyright issues that may arise from mimicking the training data (Franceschelli and Musolesi, 2021; Vyas et al., 2023). For example, BloombergGPT (Wu et al., 2023) is a 50-billion parameter auto-regressive language model that was trained from scratch on financial data sources, and exhibits superior performance on tasks in financial NLP.

While all of this progress seems set to usher in an era where companies deploy custom LLMs trained on their proprietary customer data, one of the main technical barriers that remains is *privacy*. Recent work has shown tha language model's have the potential to memorize significant swathes of their training data, which means that they can regurgitate potentially private information present in their training data if deployed to end users (Carlini et al., 2021; Jagannatha et al., 2021). LLMs are trained in a single pass and do not "overfit" to the same extent as over-parameterized models, but on specific outlier points in the training set they do have loss much lower than if the point had not been included during training (Carlini et al., 2021), allowing an adversary to perform what is called a membership inference attack (Shokri et al., 2017): given access to the model and a candidate sample $x'$, the adversary can determine with high-accuracy if $x'$ is in the training set.

While prior work shows privacy is a real

---

[*]Alphabetical order; equal contribution.

[†]Senior author, email: sneel@hbs.edu for correspondence.

concern when deploying language models, the membership inference attacks used can extract specific points but perform quite poorly *on average*, leaving significant room for improvement (Yu et al., 2023; Carlini et al., 2021). At the same time, recent works studying data deletion from pre-trained LLMs (Jang et al., 2022) and studying memorization (Carlini et al., 2023a) cite the loss of a point as a determinant of whether that point has been "deleted" from the model, or conversely is "memorized." Given that loss of a point is a relatively poor indicator of training set membership for LLMs, this raises a series of tantalizing research questions we address in this work: (i) *If a point has average "loss" with respect to a language model $\theta$, does that imply it cannot be detected as a training point?* (ii) *Can we develop strong MIAs against LLMs better than existing loss-based attacks*?

We develop a new membership inference attack we call $\text{MoPe}_\theta$, based on the idea that when the model loss is localized around a training point, $\theta$ is likely to lie in a "sharper" local minima than if the point was a test point— which does not necessarily imply that the absolute level of the loss is low. Concretely, our attack perturbs the model with mean zero noise, and computes the resulting increase in log loss at candidate point $x$ (see Figure 1).

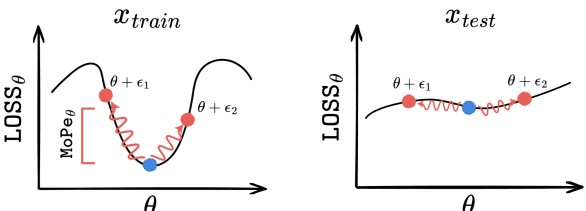

Figure 1: $\text{MoPe}_\theta$ detects training point membership by comparing the increase in loss on the training point when the model is perturbed, relative to the increase in loss of the perturbed model on a test point.

**Contributions.** In this paper we make several contributions to the developing story on language model privacy.

- We show in Section 2.2 that our $\text{MoPe}_\theta$ statistic is approximating the trace of the Hessian matrix with respect to model weights $\theta$ at $(\theta, x)$, and so can be seen as an approximation of how "sharp" the loss landscape is on $x$ around $\theta$.

- We evaluate $\text{MoPe}_\theta$ on the recently developed Pythia suite from EleutherAI (Biderman et al.,

2023b) on models that range in size from 70M to 12B. Our experimental results in Section 3 show that compared to existing attacks based on the loss ($\text{LOSS}_\theta$), $\text{MoPe}_\theta$ boasts significantly higher AUC on all model sizes up to 2.8B, and comparable AUC at 6.9B and 12B parameters. Furthermore, at low FPRs of the attack, $\text{MoPe}_\theta$ maintains high true positive rates, whereas $\text{LOSS}_\theta$ fails to outperform the random baseline (Figure 2).

- We evaluate whether the `DetectGPT` statistic proposed for detecting LLM-generated text can be repurposed as a privacy attack against pre-trained models. Our experimental results show that `DetectGPT` outperforms the loss-based attack at all model sizes up to 2.8B, but is dominated by our $\text{MoPe}_\theta$ attack at all model sizes.

- Since all the models we evaluate were trained on the same datasets in the same training order, we can make a principled study of the relationship between model sizes and order of the training points to attack success. Surprisingly, we find in Table 1 and Figure 7 that consistent with prior work $\text{LOSS}_\theta$ attack success increases with model size and proximity during training, but that these trends fail to hold for $\text{MoPe}_\theta$.

- Finally in Section 4 we train a network on MNIST (Deng, 2012) that is sufficiently small that we are able to compute the Hessian. We use this network to show that our $\text{MoPe}_\theta$ statistic does in practice approximate the Hessian trace. We also find that while $\text{MoPe}_\theta$ does outperform the random baseline, the attack performs worse than loss in this setting, highlighting that our results may be unique to language models.

These results establish our technique as state of the art for MIA against pre-trained LLMs in the white-box setting where we can access the model weights. They also challenge conventional wisdom that the loss of a point is in isolation a sufficently good proxy for training set membership.

## 2  Related Work

**DetectGPT.** The most closely related work to ours on a technical level is the recent paper

(Mitchell et al., 2023) who propose a perturbation-based method for detecting LLM-generated text using probability curvature. Their method `DetectGPT` compares the log probabilities of a candidate passage $x$ and its randomly perturbed versions $m(x)$ using the source model and another generic pre-trained mask-filling model $m$; large drops in the log probability correspond to training points. While superficially similar to our $\texttt{MoPe}_\theta$ method, there are significant differences: (i) We focus on the problem of membership inference which tries to determine if an example $x$ was present in the training data, whereas their focus is determining if $x$ was *generated by* $\theta$—an orthogonal concern from privacy and (ii) As a result, we *perturb the model $\theta$* using Gaussian noise, rather than perturbing $x$ using an additional model.

**Memorization and Forgetting in Language Models.** There are many recent works studying issues of memorization or forgetting in language models (Biderman et al., 2023a; Thakkar et al., 2020; Kharitonov et al., 2021; Zhang et al., 2021; Ippolito et al., 2023). (Jagielski et al., 2023) measures the forgetting of memorized examples over time during training, and shows that in practice points occurring earlier during training observe stronger privacy guarantees. They measure "forgetting" using membership inference attack accuracy, using an attack based on loss-thresholding with an example-calibrated threshold. By contrast (Biderman et al., 2023b) studies memorization using the Pythia suite, and finds that the location of a sequence in training does not impact whether the point was memorized, indicating that memorized points are not being forgotten over time. This finding is further confirmed in (Biderman et al., 2023a), which shows that memorization by the final model can be predicted by checking for memorization in earlier model checkpoints. Our work sheds some light on the apparent discrepancy between these two works, indicating that results in (Biderman et al., 2023a) are possible because loss value alone does not necessarily imply a point is not memorized.

**Membership Inference Attacks (MIAs).** Membership Inference Attacks or MIAs were defined by (Homer et al., 2008) for genomic data, and formulated by (Shokri et al., 2017) against ML models. In membership inference, an adversary tries to use model access as well as some outside information to determine whether a candidate point $x$ is a member of the model training set. Since (Yeom et al., 2018), MIAs typically exploit the intuition that a point $x$ is more likely to be a training point if the loss of the model evaluated at $x$ is small, although other approaches that rely on a notion of self-influence (Cohen and Giryes, 2022) or distance to the model boundary (Pawelczyk et al., 2022; Choquette-Choo et al., 2020) have been proposed. (Sablayrolles et al., 2019) argue (under strong assumptions) that the loss-thresholding attack is approximately optimal by the Neyman-Pearson lemma. The assumptions needed to show optimality do not hold in practice, however, which opens the door to substantially improving this simple attack. State of the art membership inference attacks still rely on loss-thresholding or functions of the loss (Ye et al., 2021; Carlini et al., 2022; Sablayrolles et al., 2019), but try to calibrate the threshold at which a point $x$ is declared a training point to account for the component of the loss that is specific to the example $x$. One way they do this is by (i) Training shadow models that do/don't contain the candidate point $x$, and (ii) Using the loss of these models on the point $x$ to empirically approximate the ratio of the likelihood of the observed loss $\ell(\theta, x)$ given $x$ was in the training set to the likelihood if $x$ was not (Carlini et al., 2022). A simpler, but related attack assumes access to a reference model $\theta_{ref}$ that was not trained on the point $x$ in question, and thresholds based on the ratio $\log(\ell(x, \theta)/\ell(x, \theta_{ref}))$ (Carlini et al., 2021). In this paper, we make the realistic assumption that the adversary does not have access to the computational resources to train additional language models from scratch on fresh data, or have access to a reference model that has been trained without the candidate point $x$. We note that in the case where this was possible, these same threshold calibration techniques could be used to accelerate the effectiveness of our $\texttt{MoPe}_\theta$ attack as well.

**MIAs on LLMs.** Concurrent work proposes several new MIAs tailored to LLMs. (Abascal et al., 2023) investigates MIAs where the adversary has access only to a fine-tuned model, and tries to make inferences back to the pre-trained model. Their method operates in a different access model than ours, and more importantly relies heavily on the

training of additional shadow models, and therefore does not scale to the large model sizes studied here. (Mattern et al., 2023) use a statistic similar to that used in DetectGPT (Mitchell et al., 2023) for detecting model-generated text for membership inference, in that they perturb the example $x$ using a mask-filling model (BERT) and use the resulting losses to calibrate the threshold. They conduct their experiments on the 117M parameter version of GPT-2, which they then fine-tune AG News and Twitter Data. As such their results can also be viewed as focused on small models in the fine-tuning setting, where it is known that much stronger attacks are possible (Jagielski et al., 2022; Abascal et al., 2023), in contrast to the pre-training regime studied here.

Discussion of additional related work focused on privacy issues in language models, including training data extraction, data deletion or machine unlearning, and mitigation strategies via differential privacy, are deferred to Section A of the Appendix.

## 2.1 Preliminaries

An auto-regressive language model denoted by $\theta$ takes as input a sequence of tokens $x = x_1 x_2 \ldots x_T$, and outputs a distribution $p_\theta(\cdot|x) \in \Delta(\mathcal{V})$ over a vocabulary $\mathcal{V}$, the space of tokens. Given a sequence of tokens $x$, the loss is defined as the negative log-likelihood $l$ of the sequence with respect to $\theta$:

$$\text{LOSS}_\theta(x) = - \sum_{i=1}^{T} \log(p_\theta(x_t|x_{<t}))$$

Alternatively, flipping the sign gives us the confidence $\text{conf}_\theta(x) = -\text{LOSS}_\theta(x)$. Given a training corpus $\mathcal{C}$ sampled i.i.d from a distribution $\mathcal{D}$ over sequences of tokens, $\theta$ is trained to minimize the average loss, $\frac{1}{|\mathcal{C}|} \min_\theta \sum_{x \in C} \text{LOSS}_\theta(x)$. This is typically referred to as "pre-training" on the task of next token prediction, to differentiate it from "fine-tuning" for other downstream tasks which occurs subsequent to pre-training.

**Membership Inference Attacks.** We now define a membership inference attack (MIA) against a model $\theta$. A membership inference score $\mathcal{M} : \Theta \times \mathcal{X} \to \mathcal{R}$ takes a model $\theta$, and a context $x$, and assigns it a numeric value $\mathcal{M}(x, \theta) \in \mathbb{R}$ – larger scores indicate $x$ is more likely to be a training

point ($x \in \mathcal{C}$). When equipped with a threshold $\tau$, $(\mathcal{M}, \tau)$ define a canonical membership inference attack: (i) With probability $\frac{1}{2}$ sample a random point $x$ from $\mathcal{C}$, else sample a random point $x \sim \mathcal{D}$. (ii) Use $\mathcal{M}$ to predict whether $x \in \mathcal{C}$ or $x \sim \mathcal{D}$ by computing $\mathcal{M}(x, \theta)$ and thresholding with $\tau$:

$$(\mathcal{M}, \tau)(x) = \begin{cases} \text{TRAIN} & \text{if } \mathcal{M}(x, \theta) > \tau \\ \text{NOT TRAIN} & \text{if } \mathcal{M}(x, \theta) \leq \tau \end{cases}$$
(1)

Note that by construction the marginal distribution of $x$ is $\mathcal{D}$ whether $x$ is a training point or not, and so if $\mathcal{M}$ has accuracy above $\frac{1}{2}$ must be through $\theta$ leaking information about $x$. The "random baseline" attack $(\mathcal{M}, \tau)$ that samples $\mathcal{M}(x, \theta) \sim \text{Uniform}[0, 1]$ and thresholds by $1 - \tau \in [0, 1]$ has TPR and FPR $\tau$ (where training points are considered positive). The most commonly used membership inference attack, introduced in (Yeom et al., 2018), takes $\mathcal{M}(x, \theta) = -\ell(x, \theta)$, and so it predicts points are training points if their loss is less than $-\tau$, or equivalently the confidence is greater than $\tau$. We refer to this attack throughout as the loss attack, or $\text{LOSS}_\theta$. Throughout the paper, as is common we overload notation and refer to $\mathcal{M}$ as a membership inference attack rather than specifying a specific threshold.

**MIA Evaluation Metrics.** There is some disagreement as to the proper way to evaluate MIAs. Earlier papers simply report the best possible accuracy over all thresholds $\tau$ (Shokri et al., 2017). Different values of $\tau$ correspond to tradeoffs between the FPR and TPR of the attack, and so more recently metrics like Area Under the ROC Curve (AUC) have found favor (Carlini et al., 2022; Ye et al., 2021). Given that the ability to extract a very small subset of the training data with very high confidence is an obvious privacy risk, these recent papers also advocate reporting the TPR at low FPRs, and reporting the full ROC Curve in order to compare attacks. In order to evaluate our attacks we report all of these metrics: AUC, TPR at fixed FPRs of .25 and .05, and the full ROC curves.

## 2.2 $\text{MoPe}_\theta$: Model Perturbation Attack

Our Model Perturbations Attack ($\text{MoPe}_\theta$) is based on the intuition that the loss landscape with respect to the weights should be different around training versus testing points. In particular, we expect that since $\theta \approx \text{argmin}_\Theta \sum_{x \in \mathcal{C}} \ell(x, \theta)$, where $\ell$ is the

negative log-likelihood defined in Section 2.1, then the loss around $x' \in \mathcal{C}$ should be sharper than around a random point. Formally, given a candidate point $x \sim \mathcal{D}$, we define:

$$\texttt{MoPe}_\theta(x) = \mathbb{E}_{\epsilon \sim \mathcal{N}(0, \sigma^2 I_{n_{\text{params}}})}[\ell(x, \theta + \epsilon) - \ell(x, \theta)], \quad (2)$$

where $\sigma^2$ is a variance hyperparameter that we specify beforehand, and $\theta \in \mathbb{R}^{n_{\text{params}}}$. In practice, rather than computing this expectation, we sample $n$ noise values $\epsilon_i \sim N(0, \sigma^2 I_{n_{\text{params}}})$ and compute the empirical $\texttt{MoPe}_\theta^n(x) = \frac{1}{n}\sum_{i=1}^n [\ell(x, \theta + \epsilon_i) - \ell(x, \theta)]$. This gives rise to the natural $\texttt{MoPe}_\theta$ thresholding attack, with $\mathcal{M}(x, \theta) = \texttt{MoPe}_\theta(x)$ in Equation 1. Note that computing each perturbed model takes time $O(n_{\text{params}})$ and so we typically take $n \le 20$ for computational reasons. We now provide some theoretical grounding for our method.

**Connection to the Hessian Matrix.** In order to derive a more intuitive expression for $\texttt{MoPe}_\theta(x)$ we start with the multivariate Taylor approximation (Königsberger, 2000):

$$\ell(\theta + \epsilon, x) = \ell(\theta, x) + \nabla_\theta \ell(\theta, x) \cdot \epsilon + \quad (3)$$

$$\frac{1}{2}\epsilon^T \mathbf{H}_x \epsilon + O(\epsilon^3) \quad (4)$$

where $\mathbf{H}_x = \nabla_\theta^2 \ell(\theta, x)$ is the Hessian of log-likelihood with respect to $\theta$ evaluated at $x$. Then assuming $\sigma^2$ is sufficiently small that $O(\epsilon^3)$ in Equation 3 is negligible, rearranging terms and taking the expectation with respect to $\epsilon$ of both sides of (3) we get:

$$\texttt{MoPe}_\theta(x) = \mathbb{E}_{\epsilon \sim N(0, \sigma^2)}[\ell(\theta + \epsilon, x) - \ell(\theta, x)] \approx$$

$$\mathbb{E}_{\epsilon \sim N(0, \sigma^2)}[\frac{1}{2}\epsilon^T \mathbf{H}_x \epsilon] = \frac{\sigma^2}{2}\text{Tr}(\mathbf{H}_x),$$

where the last identity is known as the Hutchinson Trace Estimator (Hutchinson, 1989). This derivation sheds some light on the importance of picking an appropriate value of $\sigma$. We need it to be small enough so that the Taylor approximation holds in Equation 3, but large enough that we have enough precision to actually distinguish the difference in the log likelihoods in Equation 2. Empirically, we find that $\sigma = 0.005$ works well across all model sizes, and we don't observe a significant trend on the optimal value of $\sigma$ (Table 2).

# 3 MIA Against Pre-trained LLMs

In this section we conduct a thorough empirical evaluation of $\texttt{MoPe}_\theta$, focusing on attacks against pre-trained language models from the Pythia suite. We show that in terms of AUC $\texttt{MoPe}_\theta$ significantly outperforms loss thresholding ($\texttt{LOSS}_\theta$) at model sizes up to 2.8B and can be combined with $\texttt{LOSS}_\theta$ to outperform at the 6.9B and 12B models (Figure 4). We also implement an MIA based on DetectGPT, which outperforms loss in terms of AUC up to size 1.4B, but is consistently worse than $\texttt{MoPe}_\theta$ at every model size. Our most striking finding is that if we focus on the metric of TPRs at low FPRs, which state-of-the-art work on MIAs argue is the most meaningful metric (Carlini et al., 2022; Ye et al., 2021), $\texttt{MoPe}_\theta$ exhibits superior performance at all model sizes (Figure 2). $\texttt{MoPe}_\theta$ is the only attack that is able to convincingly outperform random guessing while driving the attack FPR $\le 50\%$ (which is still a very high FPR!) at all model sizes.

**Dataset and Models.** We identified EleutherAI's *Pythia* (Biderman et al., 2023b) suite of models as the prime candidate for studying membership inference attacks. We defer a more full discussion of this choice to Section B in the Appendix, but provide some explanation here as well. Pythia was selected on the basis that the models are available via the open source provider Hugging Face, cover a range of sizes from 70M to 12B, and have a modern decoder-only transformer-based architecture similar to GPT-3 with some modifications (Brown et al., 2020; Biderman et al., 2023b). Models in the Pythia suite are trained on the Pile (Gao et al., 2021) dataset, which after de-duplication contains 207B tokens from 22 primarily academic sources. De-duplication is important as it has been proposed as one effective defense against memorization in language models, and so the fact that our attacks succeed on models trained on de-duplicated data makes them significantly more compelling (Kandpal et al., 2022; Lee et al., 2021). Crucially for our MIA evaluations, the Pile has clean training vs. test splits, as well as saved model checkpoints that allow us to ensure that we use model checkpoints at each size that correspond to one full pass over the dataset.

We evaluate all attacks using 1000 points sampled randomly from the training data and 1000 points from the test data. Since the $\texttt{MoPe}_\theta$

and `DetectGPT` attacks are approximating an expectation, we expect these attacks to be more accurate if we consider more models, but we also need to balance computational considerations. For `MoPe`$_\theta$ we use $n = 20$ perturbed models for all Pythia models with $\leq 2.8$B parameters, and 10 perturbed models for the 6.9B and 12B parameter models. For `DetectGPT` we use $n = 10$ perturbations throughout rather than the 100 used in (Mitchell et al., 2023) in order to minimize computation time, which can be significant at 12B.

We found the best `MoPe`$_\theta$ noise level $\sigma$ for the models of size $\leq 2.8$B parameters by conducting a grid search over $\sigma = [.001, .005, .01, .05]$. The highest AUC was achieved at .01 for the 1B parameter model, .001 for the 2.8B model and .005 for the remaining models. This suggests that there is no relationship between the optimal value of $\sigma$ and the model size. For the 6.9B and 12B parameter models, we chose a noise level of .005. We record the AUCs achieved at different noise levels in Table 2 in the Appendix. For `DetectGPT` we follow (Mitchell et al., 2023) and use T5-small (Raffel et al., 2019) as our mask-filling model, where we mask 15% of the tokens in spans of 2 words.

**Attack Success.** The results in Table 1 show that thresholding based on `MoPe`$_\theta$ dramatically improves attack success relative to thresholding based on `LOSS`$_\theta$ or `DetectGPT`. The difference is most pronounced at model sizes 160M, 410M parameters, where the AUC for `MoPe`$_\theta$ is $\approx 27\%$ higher than the AUC of .51 for `LOSS`$_\theta$ and `DetectGPT`, which barely outperform random guessing. This out-performance continues up to size 2.8B, after which point all attacks achieve AUC in the range of $.50 - .53$.

Inspecting the shape of the ROC curves in Figure 2 there is an even more striking difference between the attacks: All of the curves for the `LOSS`$_\theta$ and `DetectGPT`-based attacks drop below the dotted line (random baseline) between FPR = .5–.6. This means that even at FPRs higher than .5, these attacks have TPR worse than random guessing! This establishes that, consistent with prior work, `LOSS`$_\theta$ based attacks against LLMs trained with a single pass work extremely poorly on *average*, although they can identify specific memorized points (Carlini et al., 2022).

By contrast, the `MoPe`$_\theta$ curves lie about the dotted line (outperforming random guessing) for FPRs

that are much much smaller. Table 1 reports the TPR at FPR = .25, .05. We see that even at moderately large FPR = .25, only `MoPe`$_\theta$ achieves TPR $> .25$ at all model sizes, with TPRs $1.2-2.5\times$ higher than `DetectGPT` and `MoPe`$_\theta$. We note that none of the attacks consistently achieve TPR better than the baseline at FPR = .05, but that `MoPe`$_\theta$ still performs better than the other attacks.

| Model | Method | AUC | TPR$_{.25}$ | TPR$_{.05}$ |
|-------|--------|-----|------|------|
| 70M | LOSS$_\theta$ | .507 | .168 | .025 |
| 160M | LOSS$_\theta$ | .512 | .170 | .030 |
| 410M | LOSS$_\theta$ | .513 | .167 | .028 |
| 1B | LOSS$_\theta$ | .516 | .172 | .029 |
| 1.4B | LOSS$_\theta$ | .517 | .176 | .029 |
| 2.8B | LOSS$_\theta$ | .504 | .161 | .021 |
| 6.9B | LOSS$_\theta$ | .523 | .181 | .026 |
| 12B | LOSS$_\theta$ | .525 | .188 | .024 |
| 70M | DetectGPT | .521 | .201 | .035 |
| 160M | DetectGPT | .515 | .202 | .033 |
| 410M | DetectGPT | .525 | .206 | .032 |
| 1B | DetectGPT | .532 | .207 | .031 |
| 1.4B | DetectGPT | .534 | .210 | .029 |
| 2.8B | DetectGPT | .510 | .183 | .022 |
| 6.9B | DetectGPT | .506 | .178 | .024 |
| 12B | DetectGPT | .504 | .17 | .020 |
| 70M | MoPe$_\theta$ | .612 | .306 | .049 |
| 160M | MoPe$_\theta$ | .646 | .376 | .049 |
| 410M | MoPe$_\theta$ | .650 | .411 | .072 |
| 1B | MoPe$_\theta$ | .567 | .261 | .046 |
| 1.4B | MoPe$_\theta$ | .571 | .259 | .047 |
| 2.8B | MoPe$_\theta$ | .565 | .280 | .040 |
| 6.9B | MoPe$_\theta$ | .522 | .252 | .020 |
| 12B | MoPe$_\theta$ | .516 | .257 | .024 |

Table 1: For each model size and attack, we report the AUC, TPR at FPR .25, and TPR at FPR .05.

**Model Size.** Recent work (Carlini et al., 2023a) on the GPT-Neo (Black et al., 2021) models evaluated on the Pile has shown that as model size increases so does memorization. The data in Table 1 support this conclusion, as with increasing model size we see a (small) monotonic increase in the AUC achieved by the `LOSS`$_\theta$ attack. Interestingly, we observe almost an opposite trend in curves for `MoPe`$_\theta$, with the highest values of AUC actually coming at the three smallest model sizes! `DetectGPT` has AUC that is also flat or slightly decreasing with increased model size. We note that while this does not directly contradict prior

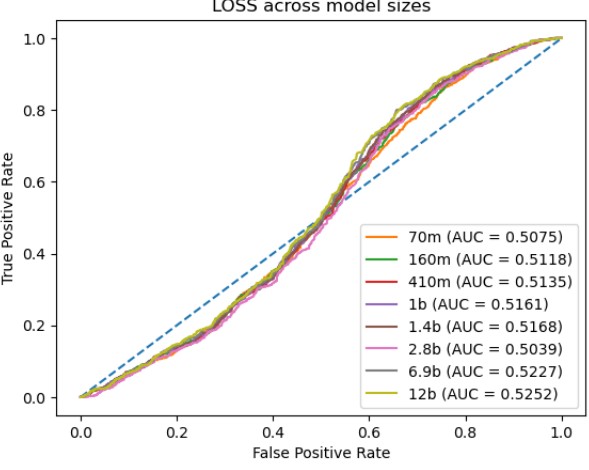

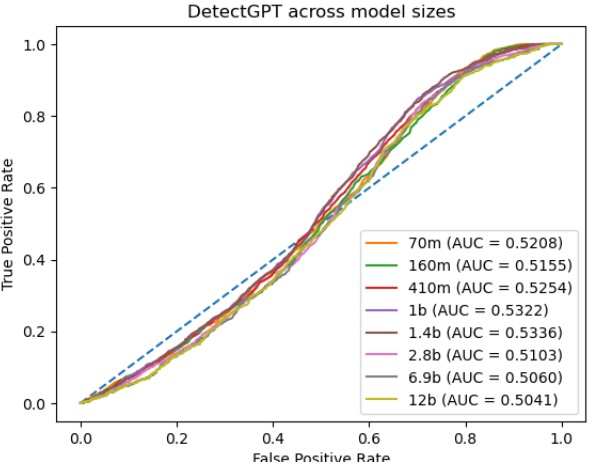

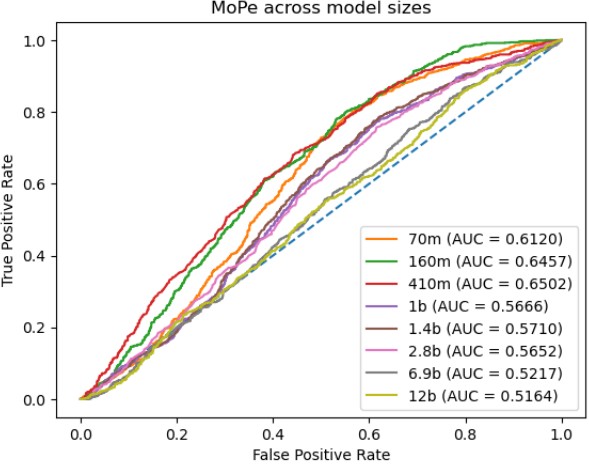

Figure 2: $\text{LOSS}_\theta$, DetectGPT, and $\text{MoPe}_\theta$ ROC Curves across all model sizes. Only $\text{MoPe}_\theta$ outperforms the baseline at FPR < .5.

results from (Carlini et al., 2023a) which use a definition of memorization based on extraction via sampling, it is surprising given the intuition from prior work. One potential explanation could be that the attack success of perturbation-based methods like $\text{MoPe}_\theta$ and DetectGPT is actually

partially inversely correlated with model size, due to variance of the Hutchinson Trace Estimator in Equation 3 increasing for larger models, and may not reflect the fact that larger Pythia suite models are actually more private.

**Training Order.** In this section we study the degree to which the models in the Pythia suite exhibit "forgetting" over time, as measured by the change in $\text{LOSS}_\theta$ and $\text{MoPe}_\theta$ statistics during training. (Jagielski et al., 2023) show that, particularly on large data sets (like the Pile for example), MI accuracy increases (equivalently loss decreases) for points in more recent batches. In Figure 7 in the Appendix, we investigate how $\text{LOSS}_\theta$ and $\text{MoPe}_\theta$ statistics vary on average depending on when they are processed during training. Concretely, we sample 2000 points from 10 paired batches $\{0 - 1, 9999 - 1e4, 19999 - 2e4, \ldots 89999 - 9e4, 97999 - 9.8e4\}$, which approximately correspond to the first and last data batches that the Pythia models see during pre-training. For each set of 2000 points, we compute the average $\text{LOSS}_\theta$ and $\text{MoPe}_\theta$ values with respect to the $\theta$ reached at the end of the first epoch. We see that, consistent with findings in (Jagielski et al., 2023), loss declines for more recent batches, but by contrast, there is no such observable pattern at any fixed model size for the $\text{MoPe}_\theta$ statistic! This finding is consistent with recent work (Biderman et al., 2023a,b) that study memorization in the Pythia suite, and find no correlation between order in training and if a point is "memorized" (as defined in terms of extractability).

**$\text{MoPe}_\theta$ vs. $\text{LOSS}_\theta$ comparison.** The disparities in MIA performance between $\text{MoPe}_\theta$ and $\text{LOSS}_\theta$ attacks shown in Figure 2 implies that there *must exist* a number of training points where the $\text{MoPe}_\theta$ and $\text{LOSS}_\theta$ statistics take very different values. The fact that $\text{MoPe}_\theta$ outperforms $\text{LOSS}_\theta$, particularly at smaller model sizes, implies that there are points with average loss values but outlier $\text{MoPe}_\theta$ values. We visualize this for the 12B parameter model in Figure 3, and include plots for all model sizes in Figure 8 in the Appendix.

**$\text{LOSS}_\theta$ and $\text{MoPe}_\theta$ Ensemble Attack.** Recall that a point is more likely to be a training point if it has a high $\text{MoPe}_\theta$ value, or a low $\text{LOSS}_\theta$ value. The above scatterplot shows that there are training points that have average $\text{LOSS}_\theta$ values and high $\text{MoPe}_\theta$ values and are easily identified by the $\text{MoPe}_\theta$ attack as

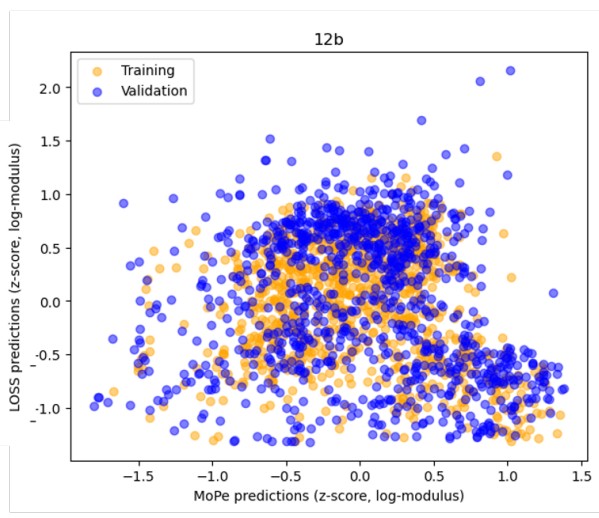

Figure 3: $-\text{MoPe}_\theta$ vs. $\text{LOSS}_\theta$ scatter plot, at model size 12B. We $z$-score the $\text{LOSS}_\theta$ and $\text{MoPe}_\theta$ values and apply a log-modulus transform $f(x) = \text{sign}(x)\log(|x| + 1)$ (John and Draper, 1980) to the scores for visual clarity.

training points but not by $\text{LOSS}_\theta$, and vice versa. This raises the obvious question of if the attacks can be combined to yield stronger attacks than either attack in isolation. We find that while for the smaller model sizes, we don't get a significant improvement over $\text{MoPe}_\theta$ in AUC by ensembling, for the two largest model sizes, where both $\text{LOSS}_\theta$ and $\text{MoPe}_\theta$ perform relatively poorly, we do get a significant improvement by thresholding on a weighted sum of the two statistics (after z-scoring). We plot the ROC curves for $\text{LOSS}_\theta$, $\text{MoPe}_\theta$, and the optimal ensemble (picked to optimize AUC) in Figure 4. At 6.9B both $\text{LOSS}_\theta$ and $\text{MoPe}_\theta$ achieve AUC = .52, while the ensemble has AUC = .55. At 12B AUC jumps from $\approx$ .52 to .544 in the ensemble.

## 4 $\text{MoPe}_\theta$ on MNIST

In this section we run our $\text{MoPe}_\theta$ attack on a small network trained to classify images from MNIST (Deng, 2012), which allows us to evaluate if the $\text{MoPe}_\theta$ statistic actually approximates the Hessian trace in practice as the derivation in Section 2.2 suggests. To test this, we sample 5000 training points and 5000 test points from the MNIST dataset, a large database of images of handwritten digits (Deng, 2012). Using a batch size of 8, we train a fully connected 2 layer MLP with 20 and 10 nodes, using the Adam optimizer with a learning rate of 0.004 and momentum of 0.9, until we reach 94% train accuracy and 84% test accuracy, a 10 percent train-test gap similar to that observed in

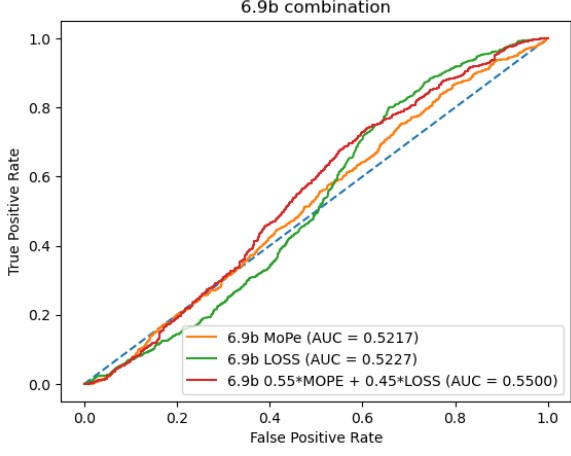

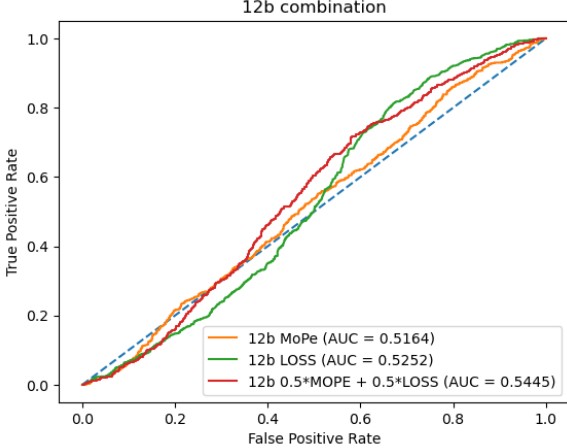

Figure 4: $\text{LOSS}_\theta$ and $\text{MoPe}_\theta$ Ensemble Attack

pre-trained language models (Carlini et al., 2021). In Figure 5 we show the distribution of the Hessian trace and scaled $\text{MoPe}_\theta$ evaluated on all training batches. The resulting distributions on training and test points are close, with a correlation between point level traces and scaled $\text{MoPe}_\theta$ values of $0.42$.

Given the generality of the proposed $\text{MoPe}_\theta$ attack, it is also natural to ask if $\text{MoPe}_\theta$ can be applied to other machine learning models. On our MNIST network, we run the $\text{MoPe}_\theta$ attack with $40$ perturbations and $\sigma = 0.05$, calculating the ROC curves in Figure 6 using 3000 train and test points. We find that $\text{LOSS}_\theta$ outperforms $\text{MoPe}_\theta$, with $\text{MoPe}_\theta$ achieving an AUC of $0.635$ and $\text{LOSS}_\theta$ achieving an AUC of $0.783$. While it does not outperform $\text{LOSS}_\theta$, $\text{MoPe}_\theta$ easily beats the random baseline, which warrants further study in settings beyond language models.

## 5 Discussion

The main finding in this work is that perturbation-based attacks that approximate the curvature in the

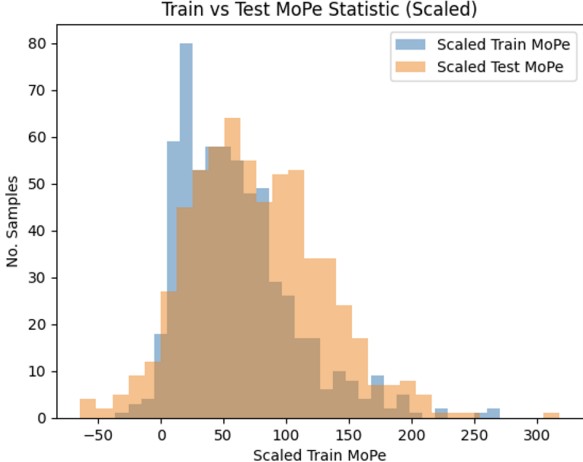

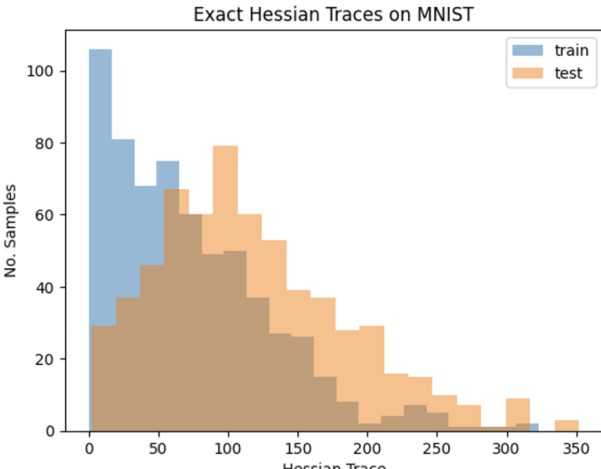

Figure 5: The distribution of MoPe statistics (left) and exact Hessian traces (right) for the MNIST dataset.

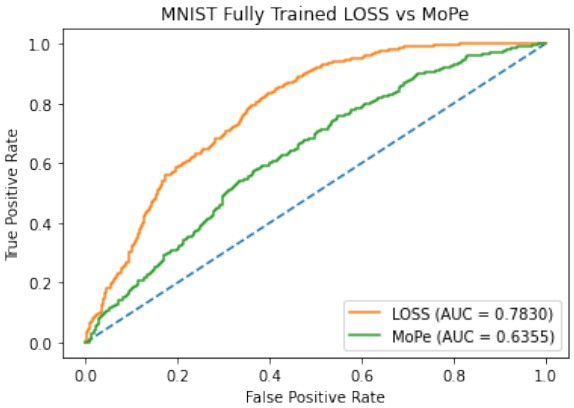

Figure 6: ROC curve for $\text{LOSS}_\theta$ vs $\text{MoPe}_\theta$ on MNIST model.

loss landscape around a given model and point, seem to perform better than attacks that simply threshold the loss. A key nuance here, is that for computational reasons, all of the attacks studied here apply a uniform threshold $\tau$ that *does not*

*depend on the point $x$.* Prior work on MIAs shows that calibrating the threshold $\tau_x$ to a specific point $x$ greatly increases attack success, for the obvious reason that certain types of points have higher or lower relative loss values regardless of their inclusion in training. Thresholding based on a loss ratio with another language model or approximating the likelihood ratio (LiRA) can all be viewed as different ways of calibrating an example-specific threshold. In this context, the results in this paper can be viewed as showing that when forced to pick a uniform threshold for an MIA score, statistics like $\text{MoPe}_\theta$ or DetectGPT that approximate the local curvature transfer better across different points than the loss. As such, methods that try to efficiently approximate an example-specific threshold, like applying LiRA or loss-ratio thresholding in the fine-tuning regime where they are more computationally feasible, or applying the very recent work of (Bertran et al., 2023) that uses quantile regression to approximate a likelihood-style attack without training additional models, in our setting is of great interest. We conjecture that such techniques will be needed to achieve high TPRs at very small FPRs. Another major unresolved question in this work is why $\text{MoPe}_\theta$ and DetectGPT success actually scales *inversely* with model size, which is likely a property of the method, namely increasing error in the Hutchinson trace estimator, rather larger models having improved privacy properties. Future work will explore other approximations to the Hessian trace that may scale better to large model sizes. Finally, future work could use $\text{MoPe}_\theta$ as part of a training data extraction attack in the style of (Carlini et al., 2021; Yu et al., 2023; Carlini et al., 2023b).

## Limitations

We now list several limitations of this work, all of which present opportunities for future research.

- Our MoPe$_\theta$ MIA operates in the white-box setting that assumes access to the model parameters $\theta$, whereas existing attacks are typically in the black-box setting, where only access to the model outputs or the loss are assumed. Nevertheless, our attack is still practical given that open source models or even proprietary models are often either published or leaked, and from a security perspective it makes sense to assume attackers could gain access to model parameters. Moreover, the findings in this paper have scientific implications for the study of memorization and privacy in language models that are orthogonal to the consideration of attack feasibility.

- We are only able to test our techniques on model sizes present in the Pythia suite (up to 12B parameters), and so the question of whether these results will scale to larger model sizes is left open.

- We were only able to optimize over a limited set of noise values in MoPe$_\theta$ due to computational reasons, and so MoPe$_\theta$ may perform even better with a more exhaustive hyper-parameter search.

- Another drawback of MoPe$_\theta$ as a practical attack, is that computing perturbed models can be computationally expensive at large model sizes, potentially limiting our ability to take enough models to accurately estimate the Hessian trace.

## Ethics Statement

We are satisfied this paper has been produced and written in an ethical manner. While the purpose of this paper is to demonstrate the feasibility of privacy attacks on large language models, we did not expose any private data in our exposition or experiments. Moreover, all attacks were carried out on open source models that were trained on *public data*, and as a result, there was limited risk of any exposure of confidential data to begin with. Finally, we propose these attacks in the hope they will spur further research on improving privacy in

language models, and on privacy risk mitigation, rather than used to exploit existing systems.

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

# A  Additional Related Work on LLM Privacy.

(Carlini et al., 2021; Yu et al., 2023) focus on the problem of *training data extraction* from LLMs by generating samples from the model, using loss-based MIAs to determine if the generated point is actually a member of the training set that has been memorized. Both papers focus more on extraction than explicitly evaluating membership inference attack success, and acknowledge existing MIAs against LLMs are relatively weak. (Jang et al., 2022) studies the problem of *unlearning* a training point from a trained model via taking gradient ascent steps. One metric they use to determine if a point has been unlearned is if the loss on a point $x$ that has been unlearned is close to the expected loss for a test point. Our work has implications for this kind of definition of unlearning, as our results show that an average $\text{LOSS}_\theta$ value does not mean the point cannot be easily detected as a training point. Differentially private training (Dwork et al., 2016) is a canonical defense against MIAs, and there has been a flurry of recent work on private model training in NLP (Anil et al., 2021a; Majmudar et al., 2022; Dupuy et al., 2022). While (Li et al., 2022) report success in *fine-tuning* language models with differential privacy, it is know that privacy during pre-training comes at a great cost to accuracy (Anil et al., 2021b). Since pre-training with differential privacy remains a challenge, existing work does not provide theoretical mitigation guarantees against our attacks on pre-trained models.

# B  Pythia Suite.

We identified EleutherAI's *Pythia* (Biderman et al., 2023b) suite of models as the prime candidate for studying membership inference attacks. Models in the Pythia suite are trained on the Pile (Gao et al., 2021) dataset, which is an 825GB dataset of about 300B tokens, consisting of 22 primarily academic sources. All our experiments are using models trained on a version of the Pile that was de-duplicated using MinHashLSH with a threshold of 0.87, which reduces the size to 207B tokens. We perform our experiments in the de-duplicated regime as it has been shown that the presence of duplicated data greatly increases the likelihood of training data memorization (Lee et al., 2021), and so attacks in the de-duplicated setting are significantly more compelling. We use a model checkpoint corresponding to one full pass over the

de-duplicated Pile. The data is tokenized using a BPE tokenizer developed specifically on the Pile. Training examples are 2048 tokens, and the batch size used during training is 1024. In order to maintain an apples-to-apples comparison between train and test examples, we batch test examples identically when evaluating our MIAs. Importantly, the Pile contains train vs. test splits which allow us to evaluate our MIAs, and is also annotated with the order of points during the training of all models which allows us to study the implications of training order for privacy.

The models in the Pythia suite are open source and available through Hugging Face, have publicly available model checkpoints saved during training, and range in size from 70m parameters to 12B. The models follow the transformer-based architecture in (Brown et al., 2020), with some small modifications (Biderman et al., 2023b).

## C    Figure & Tables

| Model | $\sigma = .001$ | $\sigma = .005$ | $\sigma = .01$ | $\sigma = .05$ |
|-------|--------|--------|--------|--------|
| 70M   | 0.6034 | 0.6069 | 0.5708 | 0.4906 |
| 160M  | 0.6394 | 0.6478 | 0.5613 | 0.5121 |
| 410M  | 0.5915 | 0.5958 | 0.5367 | 0.5190 |
| 1B    | 0.5028 | 0.5142 | 0.5924 | 0.5111 |
| 1.4B  | 0.5652 | 0.5656 | 0.5502 | 0.5136 |
| 2.8B  | 0.5320 | 0.5086 | 0.5109 | 0.5030 |

Table 2: $\mathtt{MoPe}_\theta$ AUC per model size and noise level $\sigma$.

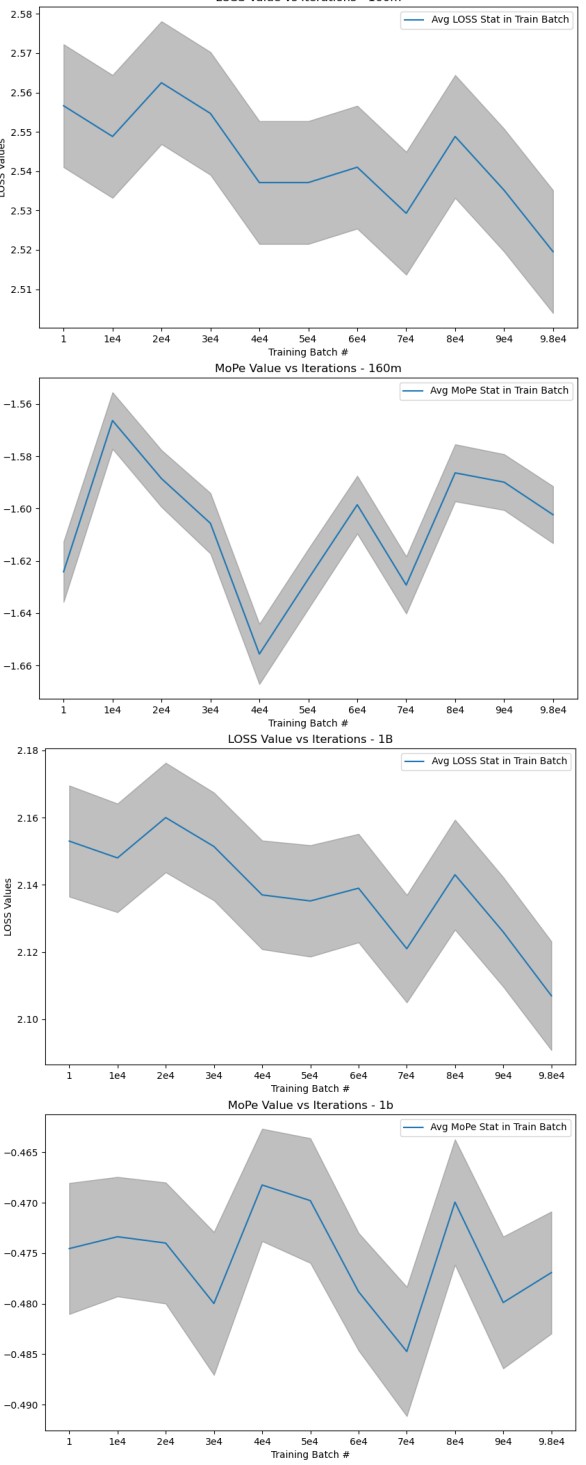

Figure 7: For each batch size, we report the average $\mathtt{LOSS}_\theta$ or $\mathtt{MoPe}_\theta$ score over points in that batch, along with a 95% CI for the average value for the 160M, 1B models.

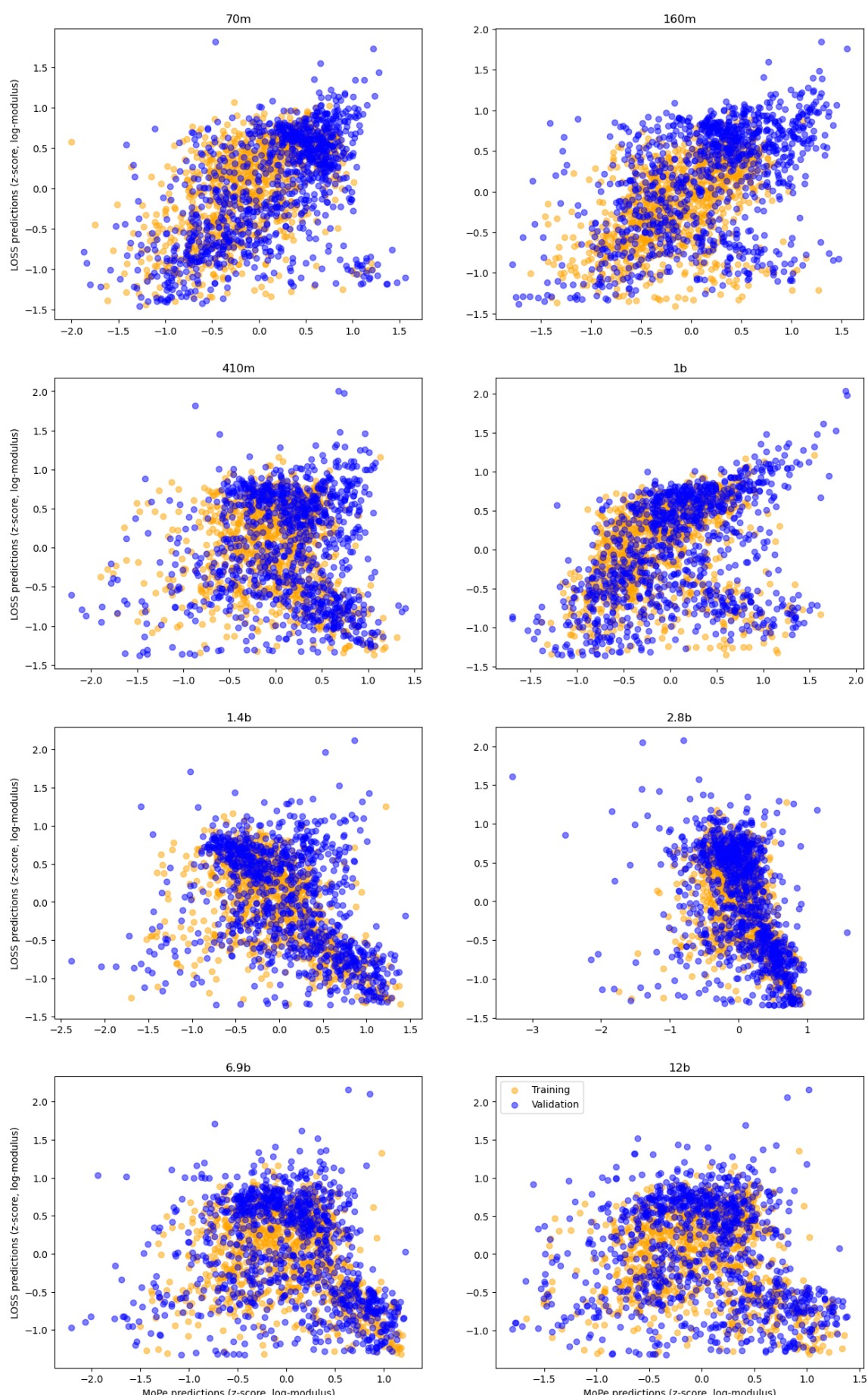

Figure 8: $\text{LOSS}_\theta$ vs. $\text{MoPe}_\theta$ scatter plots across model sizes.