# OpenReview forum: "MoPe: Model Perturbation based Privacy Attacks on Language Models"
_EMNLP/2023/Conference — EMNLP 2023 Main_

### Official Review · Reviewer_Dn8Y · 2023-08-04

**Soundness:** 3

**Excitement:**

3: Ambivalent: It has merits (e.g., it reports state-of-the-art results, the idea is nice), but there are key weaknesses (e.g., it describes incremental work), and it can significantly benefit from another round of revision. However, I won't object to accepting it if my co-reviewers champion it.

**Paper Topic And Main Contributions:**

This paper is about a membership inference attack called Model Perturbations Attack (MoPe) that can identify with high confidence if a given text is in the training data of a pre-trained language model, given whitebox access to the model's parameters. The paper shows that MoPe has improved accuracy over existing loss-based attacks and examines the role of training point order and model size in attack success.

The main contribution of this paper is the development of MoPe, which adds noise to the model in parameter space and measures the resulting drop in the log-likelihood at a given point, a statistic that approximates the trace of the Hessian matrix for model weights. The paper also shows that the loss of a point alone is insufficient to determine extractability, and there are training points that can be recovered using MoPe that have average loss. Additionally, the paper evaluates MoPe on various language models ranging from size M to B.


**Questions For The Authors:**

A. Could you discuss potential solutions or mitigation strategies to address the security risks identified in this paper?

B. The paper only considers whitebox access to the model's parameters, which may not be representative of real-world attack scenarios. How do you think the results of this study would generalize to blackbox or graybox settings?

C. The paper shows that the loss of a point alone is insufficient to determine extractability, and there are training points that can be recovered using MoPe that have average loss. Could you discuss the implications of this finding for future research on memorization or "unlearning?"

D. How do you envision this work being used in practice, and what steps can be taken to ensure that it is not misused by individuals seeking to exploit these vulnerabilities for nefarious purposes?


**Reasons To Accept:**

The strengths of this paper include the development of a novel membership inference attack that has improved accuracy over existing loss-based attacks, the examination of the role of training point order and model size in attack success, and the demonstration that MoPe accurately approximates the trace of the Hessian in practice.

If this paper were to be presented at the conference or accepted into Findings, the main benefits to the NLP community would be a better understanding of the potential security risks associated with large language models and a new method for identifying whether a given text is in the training data of a pre-trained language model. This could lead to improved security measures for language models and better protection of sensitive information.

**Reasons To Reject:**

One potential weakness of this paper is that it focuses solely on the development and evaluation of a membership inference attack, without discussing potential solutions or mitigation strategies to address the security risks identified. Additionally, the paper only considers white-box access to the model's parameters, which may not be representative of real-world attack scenarios.

**Reproducibility:**

3: Could reproduce the results with some difficulty. The settings of parameters are underspecified or subjectively determined; the training/evaluation data are not widely available.

**Reviewer Confidence:**

3: Pretty sure, but there's a chance I missed something. Although I have a good feel for this area in general, I did not carefully check the paper's details, e.g., the math, experimental design, or novelty.

---

> ### Author Rebuttal · Authors · 2023-08-28
>
> **Question 1**. Could you discuss potential solutions or mitigation strategies to address the security risks identified in this paper?
>
> **Response 1**: To our knowledge, there are no current defense strategies against MIAs against pre-trained language models; particularly since until this work, the privacy risks seemed limited. This is a good suggestion, and an exciting direction for future work that we are actively exploring, but would constitute a major paper in its own right. We discuss this in the last paragraph of the conclusion:
> "Differentially private training (Dwork et al., 2016) is a canonical defense against MIAs, and there has been a flurry of 880 recent work on private model training in NLP (Anil et al., 2021a; Majmudar et al., 2022; Dupuy et al., 2022). While (Li et al., 2022) report success in fine-tuning language models with differential privacy, it is know that privacy during pre-training comes at a great cost to accuracy (Anil et al., 2021b). Since pre-training with differential privacy remains a challenge, no existing work provides theoretical mitigation guarantees against our attacks on pre-trained models.''
>
> On a more practical note, maintaining control over the weights of the LLM prevents our MoPE attack. However, as we've noted above, and again in our next response, assuming that the adversary can't access the model weights is a brittle assumption that may not hold in practice.
>
> **Question 2**. The paper only considers white-box access to the model's parameters, which may not be representative of real-world attack scenarios. How do you think the results of this study would generalize to blackbox or graybox settings?
>
> **Response 2**. As we note in our Limitations section, we believe it’s worth studying white-box attacks both because “many open source or even proprietary models are often either published or leaked, and from a security perspective it makes sense to assume attackers could gain access to model parameters," but also because the success of these attacks is scientifically interesting in that it sheds light on how these models may memorize the underlying training data, in some cases without over-fitting. That being said, developing effective black-box attacks against pre-trained LLMs remains an exciting open question. Extending MoPE to the black-box setting seems challenging, because MoPe requires the ability to perturb the model directly.
>
> **Question 3**. The paper shows that the loss of a point alone is insufficient to determine extractability, and there are training points that can be recovered using MoPe that have average loss. Could you discuss the implications of this finding for future research on memorization or "unlearning?"
>
> **Response 3**. As we briefly mention in related work, “recent works studying issues of memorization or forgetting in language models […] measure ‘forgetting’ using membership inference attack accuracy, using an attack based on loss-thresholding with an example-calibrated threshold." We show in this work that it is possible for a training point to have loss that is approximately the average test loss, but to be easily identified as a training point based on an abnormal MoPe score. This implies that future work on unlearning or memorization should be wary of using simple metrics like the loss as a guarantee of either unlearning or memorization. This finding casts doubt on what the correct notion of ``memorization'' is for a machine learning model, and raises the question of if MIAs based on other metrics can perform even better than MoPe.
>
> **Question 4**. How do you envision this work being used in practice, and what steps can be taken to ensure that it is not misused by individuals seeking to exploit these vulnerabilities for nefarious purposes?
>
> **Response 4**. We envision this work (and the broader literature on MIAs) will serve as a toolkit model developers can use to assess the privacy of their models before releasing them to the public. By highlighting these vulnerabilities, we hope that it will cause model developers to be more wary of deploying models trained on sensitive data, and that our attacks will shed light on potential mitigation strategies. While we can't guarantee these attacks won't be used to compromise the privacy of real users (a critique that could be levied at any of the papers in the vast MIA literature), we believe that the benefits outweigh the potential risks, particularly as MIAs are a mild form of privacy violation relative to training data extraction for example.

---

### Official Review · Reviewer_EQtA · 2023-08-04

**Soundness:** 4

**Excitement:**

4: Strong: This paper deepens the understanding of some phenomenon or lowers the barriers to an existing research direction.

**Missing References:**

None that I'm aware of.

**Paper Topic And Main Contributions:**

The authors present a membership inference attack based on measuring expected changes in the model loss under random perturbations of the model. They show that the metric approximates the model Hessian, and present extensive experimental results demonstrating superiority of the attack over a conventional loss-based approach for a range of model sizes.


**Questions For The Authors:**

L145: our attack performs worse than loss in this setting --- any hypotheses for why would language models be unique?  Does it hold for LLMs with classification heads or that only produce embeddings, versus next-token prediction?

2.2: it's worth noting that the random model perturbations need only be computed once (with a cost of storing n model snapshots) and can then be applied to an arbitrary number of probe examples.

Just an observation that σ=0.005 seems quite large for model parameters that are close to 0. I imagine the hessian approximation breaks down if non-uniform noise is injected but I would be curious to understand if scaling sigma by the model weight impacts the attack success.

It would be valuable to have some error analysis on the quantities in table 1, eg using bootstrap sampling if running multiple experiments is too expensive.  When are the differences in performance statistically significant?

L482 Training order: clarify what model is evaluated for this experiment

Why evaluate the 12B model in figure 4 when there is little discernable difference between MoPe and LOSS performance?  Evaluating a smaller model will likely produce a more distinct scatter plot. Likewise, it's not clear that this figure easily supports the ensemble attack- there are nearly as many validation points with average loss and high MoPe scores.

Is it plausible that sampling more perturbed models will improve attack performance for larger models?  As noted above, the perturbed models need only be computed once, so the main cost is evaluation of all the models over all the probe samples.


**Reasons To Accept:**

The authors demonstrate a strong and relatively efficient approach to MIA without requiring shadow model training.

The work is timely and relevant given the explosion of LLMs trained with data of dubious provenance.


**Reasons To Reject:**

Success of the attack is really only demonstrated on auto-regressive LLMs.

Experimental comparison with shadow-training approaches would be valuable, at least for smaller models, or explain why these are not explored.

An important limitation is white box access, but this is not a deal-breaker.


**Reproducibility:**

4: Could mostly reproduce the results, but there may be some variation because of sample variance or minor variations in their interpretation of the protocol or method.

**Reviewer Confidence:**

4: Quite sure. I tried to check the important points carefully. It's unlikely, though conceivable, that I missed something that should affect my ratings.

**Typos Grammar Style And Presentation Improvements:**

Missing closed parenthesis on line 288 (x\in C)

---

> ### Author Rebuttal · Authors · 2023-08-28
>
> We thank the reviewer for their positive feedback and respond to all questions and feedback below.
>
> **Concern 1**: “Success of the attack is really only demonstrated on auto-regressive LLMs.”
>
> Response 1:  We focused on the Pythia Suite in this paper for  two primary reasons: (i) the auto-regressive decoder-only transformer-based architecture in GPT-Neo is similar to state of the art models (GPT-2,3,4 for example), and (ii) in order to evaluate our attacks we needed open-source models with available train/validation sets to test our attack; such a split is available automatically with The Pile. We were unable to find a clear split for other model types (e.g. BERT), although we acknowledge that such a dataset could be manually crafted if we re-trained from scratch. The Pythia Suite also allowed us to systematically benchmark MIAs across model sizes and model training (checkpoints) without pre-training from scratch.
>
> **Concern 2**: “Experimental comparison with shadow-training approaches would be valuable, at least for smaller models, or explain why these are not explored.”
>
> **Response 2**: While many shadow-training approaches to MIAs have been introduced, in the context of language models these attacks specifically require two steps of model training: some base model is pre-trained on a public data, and and a separate model is fine-tuned on private data is attacked. In our paper, we refer to these as the “pre-training” and “fine-tuning” regimes. We do not consider MIAs with shadow-training approaches in the pre-training setting, because they are not computationally feasible with the larger models studied in this paper, and indeed modern models are an order of magnitude larger than the largest model size studied here (12B). We focus on privacy attacks against pre-trained models that are also computationally feasible against realistically sized models.
>
> We note this distinction in the second-to-last paragraph of Section 1: “their [other work] results can also be viewed as focused on small models in the fine-tuning setting, where it is known that much stronger attacks are possible, in contrast to the pre-training regime studied here.” We will write additional comments emphasizing this distinction for camera-ready.
>
> **Concern 3**: “An important limitation is white box access, but this is not a deal-breaker.”
>
> **Response 3**: We acknowledge that white-box access is not always a realistic assumption; however, as we note in our Limitations section, “many open source or even proprietary models are often either published or leaked, and from a security perspective it makes sense to assume attackers could gain access to model parameters.” Moreover, our results in the white-box setting are interesting for reasons that are independent of the feasibility of MoPe as a privacy attack; they demonstrate that more memorization/privacy leakage of training data during pre-training may be occurring than previous studies that only considered the loss may suggest.
>
>
> **Question 1**: L145: our attack performs worse than loss in this setting --- any hypotheses for why would language models be unique? Does it hold for LLMs with classification heads or that only produce embeddings, versus next-token prediction?
>
> **Response 1**. One major reason LLMs are different to the MLP we train on MNIST is that the LLMs we study are trained with only one epoch, and so there is less over-fitting, which is equivalent to saying the loss-based MIA has lower accuracy. The greater diversity of sentences in the Pile relative to the images in MNIST may also contribute to the lack of over-fitting in the language models.  We will add these points to the discussion in Section 4.
>
> It would definitely be interesting to test our MIAs  on different types of LLMs, but we focused on the Pythia suite for the reasons discussed above, and because pre-trained decoder-only LLMs constitute a fundamental and important class of LLMs that typically form the basis for LLMs fine-tuned for classification for example.
>
> **Question 2**. 2.2: it's worth noting that the random model perturbations need only be computed once (with a cost of storing n model snapshots) and can then be applied to an arbitrary number of probe examples.
>
> **Response 2**: Yes! This is important to note and we will add it in our camera-ready paper. This gives our method a substantial computational advantage over an attack based on perturbing inputs for example, where the perturbations need to be computed separately for every input.
>
> **Question 3**. Just an observation that σ=0.005 seems quite large for model parameters that are close to 0. I imagine the hessian approximation breaks down if non-uniform noise is injected but I would be curious to understand if scaling sigma by the model weight impacts the attack success.
>
> **Response 3.** We note that the $\sigma$ parameter is not as large as it may seem: in the `70m` deduped Pythia model, the average of the absolute value of model weights is `0.03658` (obtained by taking `param.abs().sum()` over each parameter in `model.named_parameters()`), meaning that the standard deviation of the 0-mean Gaussian noise we add is approximately `13.7%` of the average absolute model weight.
>
> The reviewer raises an interesting theoretical point about weighting the noise levels inversely to the model size. Weighing the noise levels by the model weight size would be equivalent to weighing the partials $\frac{\partial L^2}{\partial w_i \partial w_j}$ by the magnitude of the weights $|w_i|$ in our derivation of how MoPe relates to the Hutchinson trace estimator. We did not opt for these kind of approaches because (i) we found no empirical relationship between model size and the optimal noise level (Table 2 Appendix), and (ii) it doesn't seem to have a theoretical interpretation like our Hessian trace approximation.
>
> **Question 4.** It would be valuable to have some error analysis on the quantities in table 1, eg using bootstrap sampling if running multiple experiments is too expensive. When are the differences in performance statistically significant?
>
> **Response 4.** This is a great suggestion! As suggested, we’ve performed the following procedure: using our reported values, we’ve run a bootstrap on the AUC for 9999 iterations, computed a bootstrapped SE and 95% confidence interval, and because the bootstrapped AUC distributions look approximately normal, ran a 2-sided Welch’s t-test to get p-values. Across MoPe and LOSS bootstraps, we find a SE of `0.008` to `0.013` across the model sizes. We find statistically significant differences (`p < 1e-6`) between MoPe and LOSS AUC at all model sizes respectively, even after using the Bonferroni correction. We will incorporate these values with confidence intervals into a revised table for camera ready!
>
> **Question 5**. L482 Training order: clarify what model is evaluated for this experiment.
>
> **Response 5**: Thank you for the note. We used the `160m` model here, which is in the figure but not the text. We’ll change this for camera-ready!
>
> **Question 6**. Why evaluate the `12B` model in Figure 4 when there is little discernible difference between MoPe and LOSS performance? Evaluating a smaller model will likely produce a more distinct scatter plot. Likewise, it's not clear that this figure easily supports the ensemble attack- there are nearly as many validation points with average loss and high MoPe scores.
>
> **Response 6**. We show the 12B scatter plot because the two largest model sizes are where the ensemble actually outperforms the MoPe attack in isolation. This makes sense, because for smaller model sizes MoPe is outperforming LOSS more significantly, and so including LOSS doesn't improve accuracy on average. We include scatter plots for all model sizes in Figure 10 in the Appendix.
>
> **Question 7**. Is it plausible that sampling more perturbed models will improve attack performance for larger models? As noted above, the perturbed models need only be computed once, so the main cost is evaluation of all the models over all the probe samples.
>
> **Response 7**. The reviewer makes a great point here: there is a tradeoff between speed and attack efficacy. The Hutchinson Trace Estimator does have less variance with more samples (albeit at a rate of decline proportional to the square root of the number of samples). In our experiments, we used a smaller number of models for large models because evaluation of thousands of points for testing becomes very expensive in its own right, and scaling the number of perturbed models increasing inference time linearly. Anecdotally we did not see a significant increase in attack success beyond `10` perturbed models.

---

### Official Review · Reviewer_sLT5 · 2023-08-07

**Soundness:** 3

**Ethical Concerns:**

Yes

**Excitement:**

4: Strong: This paper deepens the understanding of some phenomenon or lowers the barriers to an existing research direction.

**Justification For Ethical Concerns:**

Work proposes privacy attacks without discussing or conjecturing on defenses. It is thus balancing
towards the negative side of societal benefits scale.

**Paper Topic And Main Contributions:**

The paper presents a method (termed "MoPE") for performing a membership inference attack
(determining whether a given instance was or was not part of the training set for a given model) on
language models. The method perturbs the model parameters (hence requires knowledge of the model
internals) and stipulates that the loss landscape around model parameter perturbations is steep for
training instances and smooth for non-training instances. Thus the attack guesses instances above
some threshold of "steepness" are training instance. The paper experimentally demonstrates that
this approach does better than thresholding purely on model loss of a target instance which is
among prior baseline approaches, at least on several (nowadays) small LLMs. Among investigations of
the method, the authors also show that for other architectures (I believe all of the language
models tested were transformers) like naive dense, the loss based approach is better than the new
proposed one. Another interesting tidbit is the AUC behaviour of loss-based vs. "MoPE" is
fundamentally different, with most of LOSS's AUC coming in at the expense of false positive rate;
with MoPE being more balanced instead. Finally, the authors make a connection between the measured
quantity and traces of Hessians but the relevance of this escapes me.

**Questions For The Authors:**

- The tests on MNIST are interesting but suggest immediate follow ups with regards to the actual
  cause of the difference in MoPE vs. LOSS performance. It could be the difference in
  architectures, the difference in input types, differences in sizes, otherwise? There do exist
  some vision transformers so a good test here would be to evaluate MoPE on one. Also, less trivial
  (than dense) but standard vision architectures might also be good to test with. Dense models, as
  per the name even, are dense and could be particularly prone to memorization in one manner
  different than more sparse models.

- What are the "assumptions needed to show optimality" mentioned on line 220 ?

- Line 427 states "This suggests that there was no relationship between the optimal value of σ and
  the model size" right after noting the various differences in optimal σ and model size. Please
  explain.

- It is unclear why Figure 6 shows distributions of values instead of direct comparisons of values.
  Why is that?

- No defenses are discussed against MoPE. Do you expect them to differ from those that attempt to
  defend against membership inference in general? Are there some defenses that could work
  particularly well against MoPE (something with messing with model parameters perhaps?
  regularization?)

**Reasons To Accept:**

+ Well written work except for the points below in the first weakness. My notes include
  "nice" in response to several well-made points throughout the paper which is not common for me.

+ Attack discovers is effective at a different set of instances than loss attacks (i.e. answers the
  points nicely made on line 96). This and the AUC behaviour comparison is suggesting that there
  may be a fundamentally different sort of signal being employed in the MoPE attack (as opposed to
  being just a better LOSS). This also means the methods can be combined to get non-trivial benefit
  as shown in the Ensemble Attack. This strength has some caveats under the second weakness below.

**Reasons To Reject:**

- The description of the principal method does not match the actual method and thus the reasoning
  behind it is misplaced and/or potentially irrelevant. From the start, the method is suggesting
  that loss landscape around training points is sharper than around random (non-training) points:

       "based on the idea that when the model loss is localized around a training point, it is
        likely to lie in sharper local minima than if the point was a test point—a previously
        unexplored fact distinct from the magnitude of the loss"

       "loss around x′ ∈ D_train should be sharper than around a random point"

  The actual method perturbs the model parameters though one would expect perturbations to points
  given the above descriptions. The initial point seemed intuitive while model perturbations are
  not. The connection between model perturbations and point perturbations are not described or
  explained.

  Suggestion: Explain why input perturbations are not done in favor of model perturbations. (if
  true) explain how model perturbation is the same as, or sufficiently similar to, input
  perturbation. Other motivations might include the difficulty of perturbing inputs of discrete
  tokens (though this can be addressed by perturbing embeddings). Finally, if input perturbations
  are possible, compare the model perturbation approach to the input perturbation one,
  experimentally.

  In addition to this point, the connection to Hessian trace is specific to the (I claim) less
  intuitive model perturbation. The benefit of making such a connection are not discussed.

  Suggestion: describe the theoretical or otherwise benefits of making the connection between model
  parameter perturbation as per MoPE and Hessian traces.

- The actual method, perturbing model parameters, might have more connections to prior methods
  other than LOSS. Perturbing a model is moving it away from the set of parameters that was
  attained with a particular training point and in same way, getting closer to a model that was
  trained without that point. There are membership methods based on comparisons of "with point" vs
  "without point" models. Cited works include [Carlini 2021]. Such methods, however, were not
  compared to MoPE experimentally nor conceptually.

  Suggestion: Include a deeper discussion on attacks based on reasoning with models known to not be
  trained on target input. Provide experimental comparisons of MoPE and such methods.

- Unexplained restrictions on target models. In particular, the authors claim that single epoch
  training is common as a justification of not experimenting with models trained with multiple
  though the discussion of checkpoints (around line 410) suggests that not only is it also common
  to train for more than one pass, these models were already readily available during
  experimentation and had to be specifically avoided.

  Suggestion: Compare MoPE to baselines (LOSS and otherwise) on models trained with multiple
  passes. Demonstrate that the benefit of MoPE over baselines is or is not associated with the
  single-pass model training regimen.

Other suggestions:

- Additionally I suggest the authors revisit their related work sections and be less quick to
  dismiss comparisons against other works due to some difference in setting that could be easily
  adopted to make a direct experimental comparison possible.

**Reproducibility:**

4: Could mostly reproduce the results, but there may be some variation because of sample variance or minor variations in their interpretation of the protocol or method.

**Reviewer Confidence:**

3: Pretty sure, but there's a chance I missed something. Although I have a good feel for this area in general, I did not carefully check the paper's details, e.g., the math, experimental design, or novelty.

**Typos Grammar Style And Presentation Improvements:**

- Different notations are used for model loss throughout the work. Additional confusion is that
  LOSS is also used as the membership inference attack based on loss but also the definition of
  modeling loss before line 273.

- Order of arguments to $ l $ is backwards on Equation (3).

- Missing ")" on line 362 after σ^2.

---

> ### Author Rebuttal · Authors · 2023-08-23
>
> We thank the reviewer for their feedback and kind words, and we will address the typos raised around notational consistency.  We will focus on the questions and weaknesses raised.
>
> **Weakness 1**: The description of the principal method does not match the actual method and thus the reasoning behind it is misplaced and/or potentially irrelevant... \
> **Response 1**: We thank the reviewer for raising these points. We believe that our current description of model perturbations vs point perturbations is correct as written and offer additional clarifications below. However, we do believe that the reviewer raises an interesting point about perturbing inputs, which we mention in our related work and will expand on below as well. The training objective is $\text{argmin } l(\theta, D)$, and so we would expect that the *model* learned during training is an approximate local minimizer, because gradient descent even on a non-convex objective converges to a local minimizer. This point is conveyed in Figure $1$. Considering loss with respect to the input $x$, which is what point perturbations would seek to exploit, we note that if $x$ is a training point and $\theta$ is the learned model, it does not necessarily imply that $\nabla_{x}l(x, \theta)$ is close to zero. That being said, the idea of perturbing the inputs and calculating the drop in loss is an interesting one that has been explored in the context of detecting model-generated content, and could be leveraged as an MIA attack. We have some preliminary results using this attack, which seems to outperform loss but perform worse than MoPe, which we could include in the camera ready results, although they may warrant a separate paper. We discuss this prior work in the DetectGPT section of the related work.
>
> **Weakness 2**: The actual method, perturbing model parameters, might have more connections to prior methods other than LOSS. Perturbing a model is moving it away from the set of parameters that was attained with a particular training point and in same way, getting closer to a model that was trained without that point. There are membership methods based on comparisons of "with point" vs "without point" models. \
> **Response 2**:  We are familiar with the membership inference attacks that try to approximate the loss as if the model was not trained on the point. The most relevant work in that direction is [1] which uses the "self-influence" of a candidate point on the loss as an MIA statistic. We did not consider this attack in this work because it requires computing an inverse Hessian vector product which is inefficient; even for the vision models studied in [1] their attack was significantly slower than baseline attacks, see their Table 3.  We will add this to our discussion of prior work. There are no obvious theoretical connections between our perturbation based model and this attack, other than the connection to the Hessian we discuss in the paper. There are MIAs that use access to a model known to not be trained on the input to calibrate the threshold for the loss attack for example. We have some preliminary work using these models, and they do improve attack success (including of MoPe), but this is not a realistic assumption for pre-trained models given the cost incurred in their training. It is a much more relevant consideration for attacks against fine-tuned models, which is not the context we're studying it in. This paper explicitly focuses on the pre-training regime. We will soften our related work section to discuss these alternative approaches as suggested.
>
> **Weakness 3**:
> the authors claim that single epoch training is common as a justification of not experimenting with models trained with multiple though the discussion of checkpoints (around line 410) suggests that not only is it also common to train for more than one pass, these models were already readily available during experimentation and had to be specifically avoided. \
> **Response 3**: It is not common to train language models for multiple epochs -- the models + dataset in question (GPT-Neo on the Pile) trained for approximately 1.5 epochs, and state of the art models like GPT-3 and Llama-2 are trained with a single epoch during pre-training. That being said, training for multiple epochs would only increase the success of our MIAs, because the model would have more information from the data. The reason we didn't do this is it would make comparisons between points messier, because we would have to account for if they had been seen once or twice by the model. We could compare to models trained on multiple passes, which we did do for the experiment on MNIST, but it would not be accurate in terms of how these language models are actually trained.
>
> We'll now answer the questions from the reviewer.
>
> **Question 1**: There do exist some vision transformers so a good test here would be to evaluate MoPE on one. Also, less trivial (than dense) but standard vision architectures might also be good to test with. Dense models, as per the name even, are dense and could be particularly prone to memorization in one manner different than more sparse models. \
> **Response 1:** We agreed that more extensive experiments with vision models, including transformers, is an interesting direction for future work. In this paper our focus was the state of the art auto-regressive language models that have become very popular lately, and so we didn't focus on a more thorough evaluation within the vision context.
>
> **Question 2**: What are the "assumptions needed to show optimality" mentioned on line 220 ? \
> **Response 2:** The assumption in Equation (1) in [2], and it is that the model output from the training algorithm is a sample from the posterior distribution in Equation (1): $\theta \sim e^{\frac{-1}{T}\sum_{i = 1}^{n}l(\theta, x_i)}$, which does not hold for non-convex objectives like the one's used to train LMs.
>
> **Question 3**: Line 427 states "This suggests that there was no relationship between the optimal value of σ and the model size" right after noting the various differences in optimal σ and model size. Please explain. \
> **Response 3:** By this we did not mean that the value of $\sigma$ didn't effect the ultimate attack success for a given model size, rather we meant that there was no monotonic trend we observed in the relationship between the optimal $\sigma$ and the model size, e.g. it was not the case that larger models corresponded to larger or smaller values of sigma. We will clarify this wording for the camera ready.
>
> **Question 4**: It is unclear why Figure 6 shows distributions of values instead of direct comparisons of values. Why is that?
> **Response 4**: Figures 5,6 compare the distribution of train vs. test points mope statistics and exact hessian traces. This is because the comparison of those two distributions is a direct determinant of MIA attack success. A related but different question, is for a given point, how does the hessian trace compare to the mope statistic (e.g. what is the approximation error for MoPe). We do report a version of this statistic (point-wise correlation of .42) but we agree it would be useful to have a plot comparing point-wise values, and we can easily add that to the supplement for the camera ready.
>
> **Question 5:** No defenses are discussed against MoPE. \
> **Response 5**: To our knowledge, there are no current defense strategies against MIAs against pre-trained language models; particularly since until this work, the privacy risks seemed limited. This is a good suggestion, and an exciting direction for future work that we are actively exploring, but would constitute a major paper in its own right. We discuss this in the last paragraph of the conclusion: \
> "Differentially private training (Dwork et al., 2016) is a canonical defense against MIAs, and there has been a flurry of 880
> recent work on private model training in NLP (Anil et al., 2021a; Majmudar et al., 2022; Dupuy et al., 2022). While (Li et al., 2022) report success in fine-tuning language models with differential privacy, it is know that privacy during pre-training comes at a great cost to accuracy (Anil et al., 2021b). Since pre-training with differential privacy remains a challenge, no existing work provides theoretical mitigation guarantees against our attacks on pre-trained models.''
>
>
> References:
> [1] https://arxiv.org/abs/2205.13680 [2] https://arxiv.org/pdf/1908.11229.pdf

---

### Meta-Review · Area_Chair_t2VS · 2023-09-23

**Recommendation:** 5

**Metareview:**

The paper presents a membership inference attack method called MoPe (Model Perturbations Attack) to detect if a certain text is in the training data of a pretrained LLMs. The proposed method assumes access to model parameters.

Pros:
* The authors demonstrate that the proposed method is very effective in identifying if a given text is in the training data with a high confidence, compared to loss-based attacks.
* The paper is well written and very timely. I believe this direction of work will lead to our improved understanding of security, safety and trust related issues with LLMs.

Cons:
* The reviewers have identified two major weaknesses in the paper: 1) The method assumes access to model parameters, and 2) the paper does not provide or discuss potential solutions or mitigation strategies to address the risks identified in this paper.

The authors have responded to both limitations adequately in their rebuttals. Despite these two limitations, the reviewers have agreed that the proposed setup is worth exploring and the paper makes a valuable contribution which will be of interest to the community and to the development of LLMs.

---

### Decision · Program_Chairs · 2023-10-07

**Decision:**

Accept-Main

**Comment:**

The paper presents a membership inference attack method called MoPe (Model Perturbations Attack) to detect if a certain text is in the training data of a pretrained LLMs. The proposed method assumes access to model parameters.

Pros:
* The authors demonstrate that the proposed method is very effective in identifying if a given text is in the training data with a high confidence, compared to loss-based attacks.
* The paper is well written and very timely. I believe this direction of work will lead to our improved understanding of security, safety and trust related issues with LLMs.

Cons:
* The reviewers have identified two major weaknesses in the paper: 1) The method assumes access to model parameters, and 2) the paper does not provide or discuss potential solutions or mitigation strategies to address the risks identified in this paper.

The authors have responded to both limitations adequately in their rebuttals. Despite these two limitations, the reviewers have agreed that the proposed setup is worth exploring and the paper makes a valuable contribution which will be of interest to the community and to the development of LLMs.